# Conditional networks for screening of breast cancer metastases in lymph nodes

**Gianluca Gerard**                GIANLUCA.GERARD01@UNIVERSITADIPAVIA.IT  and  **Marco Piastra**
MARCO.PIASTRA@UNIPV.IT
*Laboratorio di Visione Artificiale e Multimedia, Dipartimento di Ingegneria Industriale e dell'Informazione*
*Università degli Studi di Pavia, Via Ferrata 5, 27100 Pavia, ITALY*

## Abstract

We assess the viability of applying a few-shot algorithm to the segmentation of Whole Slide Images (WSI) for human histopathology. The specific field considered is finding metastatic lesions in sentinel lymph-nodes. Given the huge size of WSIs and the substantial effort required by human pathologists to analyze them for diagnostic purposes, the goal is to design a system that could perform an automatic screening by segmenting out those areas that contain elements of potential interest. 'Classical' supervised techniques have found limited applicability in this respect, since their output cannot be adapted unless through extensive retraining. The approach to segmentation of histopathological images presented here is based on conditional FCN (co-FCN) networks, in which a fully convolutional network conditioned on an annotated support set of images do inference on an unannotated query image. After a complete end-to-end training, it is possible to correct the behavior at run time of a co-FCN by extending the support set, without further optimization. The adoption of co-FCN is expected to ease the annotation task and also to improve the acceptance by human experts, who will be able to correct the co-FCN behavior incrementally. In this preliminary work we use the publicly-available Camelyon16 dataset to show that the segmentation produced by co-FCN trained using late fusion and sparse annotations can be effectively modified at runtime, by integrating corrections on the fly.

**Keywords:** fully convolutional network, few-shot learning, meta-learning, sparse annotation, lymph nodes, camelyon16, histopathological images

## 1. Introduction

Our goal is to design an automated segmentation method for the automated screening of Whole Slide Images (WSI) of sentinel lymph-nodes that detects areas which might contain metastatic lesions, for a subsequent inspection by a human pathologist. In the intentions, the method should be the basis for a more interactive and collaborative training of the system, with a substantial involvement of human pathologists in the training loop.

## 2. Methods

The method adopted is based on a particular variant of the few-shot segmentation method devised by Rakelly et al. (2018), which uses conditional FCN (co-FCN) in which a fully convolutional network (FCN) (Shelhamer et al., 2016) conditions on an annotated sup-

port set of $k$ images to perform inference on an unannotated query image. A schematic representation of the architecture is provided in Figure 1

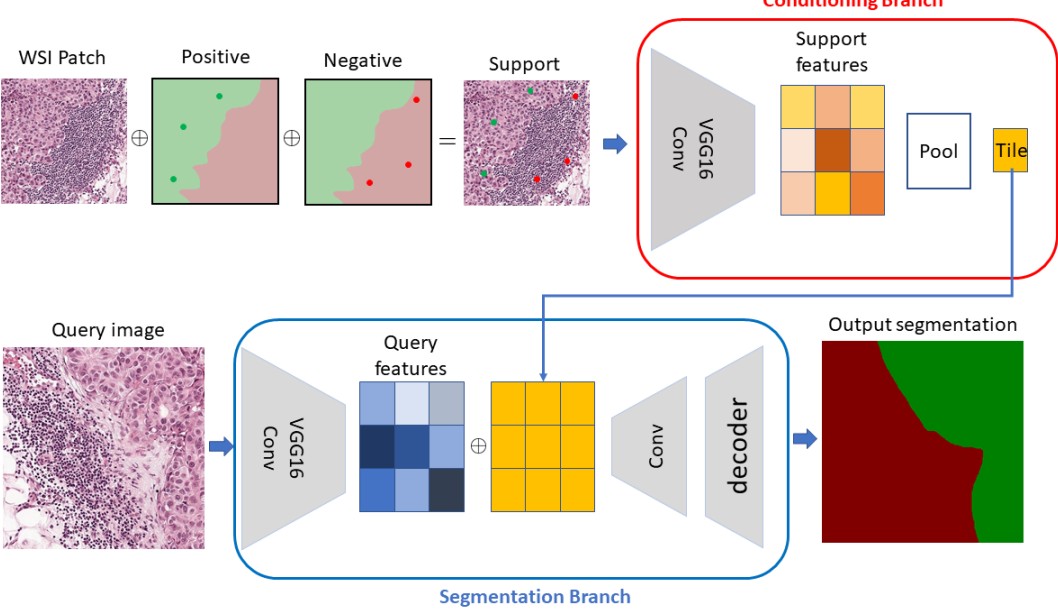

Figure 1: The architecture of co-FCN is shown here. The top branch (conditioning branch) takes as input the channel-wise concatenation of a WSI patch and its sparse or dense annotations. The bottom branch (segmentation branch) conditions on the output of the top branch to densely segment the query. The two branches are learned jointly and end-to-end.

The end-to-end training of the co-FCN is performed as follows:

- the training set contains sentinel lymph-node WSIs at 20x from the publicly-available Camelyon16 dataset(Bejnordi et al., 2017), each associated with dense annotations corresponding to two classes: lesion and non-lesion;

- WSIs are divided in patches (448x448 pixels) and the corresponding masks are generated. A VGG16 network (Simonyan and Zisserman, 2014), the backbone of the FCN, is pre-trained as a classifier onto the training set of patches: each patch is classified as either lesion or non-lesion depending on the presence of lesion pixels in the central 224x224 pixels area (Liu et al., 2017);

- pairings, i.e. "few-shot segmentation tasks" (Rakelly et al., 2018), are created in the training set, by first picking one class at random and then sampling one query image and $k$ support images with the condition that all sampled images must contain pixels from the picked class;

- in each pairing, dense annotations of the $k$ support images are converted into sparse annotations by sampling the corresponding masks with uniform distribution;

- the co-FCN is trained end-to-end onto the training set of tasks.

## 3. Test procedure

We envision that trained pathologists will guide the automatic co-FCN segmentation output via error corrections as a support set. To evaluate the performance of co-FCN in incorporating the corrections, we follow the procedure below:

1. we create new pairings, by selecting query and support patches as described in section 2 from a dataset of annotated WSIs from Camelyon16 (the query patches are sampled from WSIs that were not included in the previous training set);

2. the trained co-FCN is applied in inference mode to each pairing to produce a predicted segmentation; each predicted segmentation is compared with the ground truth to produce a four-class segmentation: lesion (TP), non-lesion (TN), false lesion (FP) and false non-lesion (FN);

3. we create new pairings, with query patches sampled from Camelyon16 WSIs not previously used and with a support set extracted from a subset of the query images segmented in the previous stage;

4. the sparse annotations of the later support set are derived from the new four semantic classes obtained previously; the trained co-FCN is used in inference mode to each pairing to produce a new predicted segmentation; we record the average accuracy and Intersection over Union (IoU) metrics obtained on the query patches;

5. the sparse annotations for the same support set patches are recreated starting from the original lesion and non-lesion semantic classes; the trained co-FCN is reused in inference mode on each pairing to produce a new set of segmentation; new accuracy and IoU metrics are computed and compared with the previous results.

## 4. Conclusions and results

We trained a co-FCN model on the Camelyon16 tumor WSIs with identifier 1 through 55 and with a 5 shots/10 points sparse annotations support set. On query images extracted from four WSIs, the accuracy and Intersection over Union (IoU) metrics obtained with the support set and the four classes sparse annotations performs significantly better on non-lesion regions (98% accuracy and 97% IoU) than with the original two classes sparse annotations (87% accuracy and 86% IoU). Within a 2% range we have comparable performance on lesions regions (43% accuracy and 38% IoU) with a slight advantage of the inference conducted with the original annotations. As a next step we will investigate the influence of different sample of support sets on the segmentation results. We will also study and test additional strategies to include the correction of errors in the training and/or inference flows.

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
