# OpenReview forum: "Conditional networks for screening of breast cancer metastases in lymph nodes"
_MIDL.io/2019/Conference/Abstract — MIDL Abstract 2019_

### Official Review · AnonReviewer2 · 2019-04-30
**Interesting application of few-shot segmentation via co-FCN in the context of whole slide images.**

**Rating:** 3
**Confidence:** 3

**Review:**

This is an interesting work in which a variant of conditional FCN is used to segment Whole Slide Images (WSI) of metastatic lesions in sentinel lymph-nodes. The proposed few-shot segmentation method is based on previous work by (Rakelly et al, 2018) and might allow expert pathologists to guide the segmentation process by interactively correcting the support set of images.
The paper is clear and the training and test procedures are well explained. I believe it constitutes an interesting application of co-FCN in the medical context that merits publication as an abstract paper at MIDL 2019.

---

### Official Review · AnonReviewer1 · 2019-04-30

**Rating:** 3
**Confidence:** 3

**Review:**

This work applies a few-shot algorithm for whole slide image segmentation in histopathology. The topic is interesting and of large significance in clinical practice. How will the location distribution of sampled support set affect the training? Meanwhile, more experimental results can be presented. I suggest the manuscript can be better organised and formatted. For example, it can be adjusted into more balanced sections.

---

### Decision · Program_Chairs · 2019-05-06
**Acceptance Decision**

Accept